# Ablative Therapy in Non-HCC Liver Malignancy

**DOI:** 10.3390/cancers15041200

**Published:** 2023-02-14

**Authors:** Tyler P. Robinson, Travis Pebror, Matthew E. Krosin, Leonidas G. Koniaris

**Affiliations:** 1Department of Surgery, Indiana University, Indianapolis, IN 46202, USA; 2Department of Interventional Radiology, Indiana University, Indianapolis, IN 46202, USA

**Keywords:** liver malignancy, radiofrequency ablation, microwave ablation, cryotherapy, photodynamic therapy, percutaneous ethanol injection, Irreversible Electroporation, high intensity focused ultrasound, stereotactic body radiotherapy, laser-induced thermotherapy, electrochemotherapy

## Abstract

**Simple Summary:**

Liver malignancy affects thousands of people, and its treatment is constantly evolving. Ablative therapies are a series of minimally invasive modalities that treat these tumors in combination with other established forms of treatment such as chemotherapy and surgery. Ablative therapy is well-studied in the case of the most common type of liver malignancy, hepatocellular carcinoma. There is much less information on ablative therapy in non-hepatocellular carcinoma liver malignancy treatment. Therefore, we have described the available literature, focusing on ablative therapy’s promising results, shortcomings, and detail areas for future research on the topic of non-hepatocellular carcinoma of the liver.

**Abstract:**

Surgical extirpation of liver tumors remains a proven approach in the management of metastatic tumors to the liver, particularly those of colorectal origin. Ablative, non-resective therapies are an increasingly attractive primary therapy for liver tumors as they are generally better tolerated and result in far less morbidity and mortality. Ablative therapies preserve greater normal liver parenchyma allowing better post-treatment liver function and are particularly appropriate for treating subsequent liver-specific tumor recurrence. This article reviews the current status of ablative therapies for non-hepatocellular liver tumors with a discussion of many of the clinically available approaches.

## 1. Introduction

In 2022 it is predicted that in the United States, there will be over 41,000 new cases of liver cancers resulting in over 30,000 deaths [1]. Thus, there is a critical need to optimize therapies for these patients to both improve quality of life and survival time, and decrease mortality. In addition to systemic therapies such as chemotherapy and immunotherapy, local therapies can offer significant palliation or impart cures with less morbidity than conventional surgical approaches. These non-surgical local therapies include a variety of approaches that cause focused destruction of the liver cancer while better preserving normal adjacent cells. Such therapies include external beam radiation, embolization, and point-source based ablations. 

Ablative therapies focus on various forms of destructive external energy within a restricted region of tissue to destroy cells within an ablative zone with little effect on adjacent healthy tissue. The first modern use of ablation was for trigeminal neuralgia in 1931 by Martin Kirschner using Radiofrequency Thermocoagulation [2]. The idea of tumor ablation subsequently has been traced back to the 1970s by Taylor who proposed the use of a hypodermic syringe to inject electromagnetic energy into tissues deep in the body [3]. Modern examples of ablative technologies directed at cancer include heat killing of cells utilizing energy from a variety of electromagnetic spectra including microwave and X-ray spectra, freezing and thawing cells, applying external voltage potentials to disrupt cell membranes, local instillation of toxins such as ethanol, and focused external ultrasound waves. 

Ablative therapies are an attractive potential therapy for patients with localized tumors, particularly of the liver where directed ablation can spare traumatic injury to the liver and preserve functional liver mass. Some ablative therapies offer clinical outcomes on par with definitive surgical treatment, while others currently function as a less effective though far less morbid alternative. This is of particular concern for patients who might not tolerate an operation due to comorbidities such as cirrhosis or require extensive resections that might be prohibitively morbid due to regenerative limitations on liver parenchyma from underlying liver disease or chemotherapy-associated liver disease [4]. The neoadjuvant and adjuvant roles of ablation in conjunction with surgical and systemic therapies are ever-expanding and may offer an alternative to systemic chemotherapeutic agents such as irinotecan and oxaliplatin which cause fatty liver disease and sinusoidal damage [4,5]. Finally, in certain circumstances ablation offers a means of palliation and can be performed in the outpatient setting, reduces morbidity and mortality, decreases cost, and allows for real-time imaging to direct therapy [6,7,8].

## 2. Review Purpose

There has been abundant research conducted on outcomes following ablative therapies in the most common type of primary liver malignancy, hepatocellular carcinoma (HCC) of the liver, however, there has been a more limited discussion of ablative therapies in non-HCC malignancies. Herein, we describe the techniques, (which are summarized in Table 1) and review the literature on ablative therapies applied to primary and metastatic non-HCC malignancies. Liver metastasis occurs in 5% of cancer patients, and in the United States, is more common than primary liver cancers [9]. After HCC, the second most common type of primary liver malignancy is cholangiocarcinoma, which worldwide is responsible for 10–20% of primary liver malignancy [10]. Primary liver tumors such as hepatoblastoma (100 new diagnoses per year in the United States [11]) and angiosarcoma (200 new diagnoses per year in the world [12]) lack data on ablative therapeutic treatment and for the purpose of this review will not be discussed.

### Materials and Methods

This research was deemed exempt by the Institutional Review Board by Indiana University. All authors completed institutional training and certification on the conduct of human subject biomedical research. An exhaustive search for articles related to various ablative technologies was undertaken by generating a list of keywords related to each candidate modality to search on publicly available databases including PubMed and Google Scholar. Candidate articles were screened for relevance to the narrow clinical focus by reviewing the abstract of each article. Articles related to hepatocellular carcinoma treatment were excluded, as were those not directly related to human subjects. A list of candidate review articles and clinical studies was generated and grouped by ablative modality. The authors reviewed the available literature and concluded that there was sufficient literature to describe the clinical application of radiofrequency ablation, microwave ablation, cryotherapy, high-intensity focused ultrasound, photodynamic therapy, stereotactic body radiotherapy, laser-induced thermotherapy, electrochemotherapy and percutaneous ethanol injection in treatment of non-hepatocellular carcinoma malignancy in the liver. Next, the institution’s Picture Archiving and Communication System (PACS) was queried for procedural images representative of each modality. Images were completely deidentified and stored within a secure database for submission. 

## 3. Impact of Non-HCC Primary Liver Malignancy on Prognosis

The 5-year survival for patients with cholangiocarcinoma is as low as 10% with treatment. Without treatment, the median survival is 3.9 months [36,37]. The median survival of patients with intrahepatic cholangiocarcinoma who undergo surgical resection is 28 months and the 5-year survival is 30% [38]. Resection is the mainstay for cholangiocarcinoma treatment, however, even after R0 resection, or a microscopically negative tumor margin, 60–70% of these patients have cancer recurrence [39]. A German retrospective multicenter study that examined survival in patients with recurrence of intrahepatic cholangiocarcinoma found that patients who underwent repeat hepatectomy for recurrence had a median overall survival of 65.2 months and 5-year overall survival of 57% [40]. Recently a multi-center prospective trial has demonstrated five-year survival for patients undergoing neoadjuvant therapy followed by liver transplantation for the treatment of nonresectable cholangiocarcinoma [41].

To date, non-resective local therapies for extra-hepatic cholangiocarcinoma remain limited to palliative-intent treatments. These have been particularly focused on photo-dynamic and laser therapies to relieve biliary obstruction in combination with stent placement [42]. Retrospective data seems to support this approach as potentially improving quality of life and survival, but supportive data for this remains small retrospective case series [43].

## 4. Impact of Metastatic Disease to the Liver on Prognosis

One of the most common sites of distant metastatic disease in solid tumors is the liver [44]. The one-year survival of all patients with liver metastasis is 15% [9]. The majority of metastasis to the liver are adenocarcinomas, however neuroendocrine tumors, squamous cell carcinomas, lymphoma, sarcoma, and melanoma also metastasize to the liver. A previous review thoroughly describes metastatic disease to the liver [45]. Organs with primary tumors that metastasize to the liver include the colon and rectum, breast, lung, pancreas, and stomach [9,45]. For metastatic disease with multiple sites in the liver, major resection is associated with worse outcomes than parenchymal-preserving smaller resections.

All patients with colorectal cancer have a 1-, 5-, and 10-year overall survival of 75%, 44%, and 33%, respectively and the median overall survival was two years for stage III-IV [46]. Due to the portal circulation and hepatic artery blood supplies of the liver, as many as 50% of patients diagnosed with colorectal cancer will develop liver metastasis [45,47]. In a large retrospective Dutch study of over 23,000 patients with liver metastasis, the most common type of metastasis was of colorectal origin, which was present in about 35% of liver metastasis [48]. At autopsy of over 5800 patients with colorectal cancer metastatic disease was found in 29% of patients. Of those with metastatic disease, metastasis to the liver was found between 32% and 73% depending on the histological subtype [49]. Patients who have colorectal liver metastasis that is untreated have a median survival of 6.9 months with a 5-year survival of less than 5% [50,51]. Patients who undergo metastasectomy have a 5-year survival of 20% to 50%, however, only 10% to 25% of patients are eligible for resection [45,52]. For those with unresectable diseases, improvements in liver transplant science have allowed for transplantation as a reasonable curative treatment. An initial prospective trial, SECA-1, demonstrated overall five-year survival of 60% in 21 patients undergoing orthotopic liver transplantation for the treatment of colorectal metastasis [53]. Based on these results, inclusion criteria for liver transplantation were refined for SECA-II which demonstrated overall survival of 83% [54]. 

When considering all patients with melanoma the 5-year survival rate is 93% [1]. Melanoma liver metastasis is found in 10% to 20% of patients with melanoma and these patients have a median survival of 4 to 28 months [45,55]. This is consistent with all patients with stage M1c melanoma who have a median overall survival of 5.1 months [56]. If patients undergo metastasectomy the median survival was found to be 24 months with a median 5-year survival of 24% [57]. At autopsy of patients with liver metastasis, 2% were from melanoma [48]. 

Small cell lung cancer metastasizes to the liver in 20% of patients and non-small cell lung cancer in 13% of patients with median survival under 1 year [58,59]. The median overall survival for patients with small cell lung cancer is 7 months, and the median overall survival for patients with non-small cell lung cancer is as low as 5 months [60,61]. The 5-year overall survival for all patients with lung cancer is 22% [1]. At autopsy of patients with liver metastasis, lung cancer was found in about 12% of patients [48]. The 5-year survival for all patients with breast cancer is 90% [1]. Breast cancer patients develop liver metastasis in up to 50% of patients and have a median survival of 3 to 15 months [45,62]. All patients with pancreas cancer have a 5-year survival of 11% [1]. Greater than 50% of patients who are diagnosed with pancreatic cancer are found to have liver metastasis at the time of diagnosis [63]. The median survival is less than 6 months regardless of treatment in these patients [45,63]. The median overall survival for neuroendocrine tumors is 9.3 years [64]. Neuroendocrine tumors are found to have liver metastasis in up to 75% of patients and patients who go untreated have a 13% to 54% 5-year survival [65]. This 5-year survival increases to 60–86% with surgery and the median survival was 125 months [66]. Of note, >90% of patients with neuroendocrine metastasis to the liver are not candidates for surgery [67]. The 5-year relative survival rate of all gastric cancer patients is 43% [68]. Gastric cancer patients present with liver metastasis in 4–14% of patients and have metachronous liver lesions after gastric resection in up to 37% of patients. The median survival was between 7 and 14.1 months [69]. For soft tissue sarcoma, the 5-year survival is 58% and the median survival is 8.2 years [70]. Soft tissue sarcoma presents with liver metastasis in 3% of patients and had a 1-year overall survival of 36% while patients who had non-liver metastasis had a 1-year overall survival of 81% [71]. Metastasis to the liver from any primary cancer has devastating effects on patient survival and subsequent treatment strategies are paramount to improving patient outcomes.

## 5. Radiofrequency Ablation

### 5.1. Technique

Radiofrequency ablation utilizes electric current alternating at high-frequency to cause the destruction of the intended tissue. The procedure is often performed under imaging guidance, using some combination of real-time ultrasound guidance with or without confirmatory computed tomography (CT) guidance (including CT Fluoroscopy, Helical CT, and/or cone-beam CT) (Figure 1) Trans biliary endoscopic approaches have also been explored and several open trials are registered at clinicaltrials.gov. Treatment of the lesion is done with monopolar energy whereby a closed circuit is formed between an electrode needle within the ablation probe and a dispersive electrode via the patient’s body [13]. As the current attempts to pass from the electrode needle through the tissue and back to the dispersive electrode, a radiofrequency generator alternates the current at a high frequency causing ion agitation, friction, and subsequently heat deposition within a defined region adjacent to the ablation probe. The tissue is heated to 50–100 degrees Celsius to induce coagulation necrosis [6,14]. Beyond 100 degrees Celsius, the delivery of energy is limited by increased tissue impedance as a result of the carbonization of tissue and water vaporization [35]. Modern RFA systems include automated methods to measure tissue impedance and adjust monopolar energy characteristics in real-time. The heat is concentrated concentrically around the tip of the electrode needle in various shapes and diameters depending on how the probe is constructed, the energy applied, and the tissue characteristics of the lesion. Larger regions of ablation are achieved by overlapping ablative zones of multiple probes with the goal of obtaining a 1 cm tumor-free margin [6,72,73,74]. In tissues adjacent to large blood vessels, a phenomenon known as “heat sink” occurs whereby blood flow causes perfusion-mediated tissue cooling prohibiting the tissue from reaching a minimum temperature for tissue necrosis [75]. This can be addressed through pre-ablation occlusion of the arterial supply of the tumor [76] either with preceding transarterial embolization, or with intra-operative temporary vessel clamping. 

### 5.2. Radiofrequency Ablation in Non-HCC Primary Liver Malignancy

Currently, surgery is the standard of care for cholangiocarcinoma, however, even after R0 reSection 60–70% of these patients have cancer recurrence and hepatectomy carries significant operative mortality of over 9% in a single center study [39,77]. In the event the patient is not a candidate for surgery, RFA might be a useful option, however given the low incidence of cholangiocarcinoma, poor overall survival after diagnosis, and existing definitive surgical therapy, data describing the role of RFA ablation in cholangiocarcinoma is limited to observation of outcomes in patients ineligible for initial resection or in cases of recurrence not amenable to reoperation. Given the lack of data, expert consensus provides some guidance including the European Association of the Study of Liver which offers guidelines for ablative therapy of cholangiocarcinoma. These guidelines suggest considering ablative therapy in small single lesions ≤3 cm in patients who are not candidates for surgery [78]. These guidelines are extrapolated from a more robust study of RFA ablation in the treatment of HCC. A small retrospective study of 13 nonsurgical candidates with less than three total lesions supported restricting RFA ablation to small lesions. This study demonstrated a technical procedural success rate of 88%. The 12% of patients with failed procedures had lesions >7 cm. RFA may not be an optimal choice in lesions this large. Overall, the 5-year survival of these nonsurgical candidates initially treated with RFA was 15% and the median survival of this group was 38.5 months [79]. In comparison, initial treatment with major hepatic resection demonstrates improved 5-year survival of 26% with a worse median survival of 27.6 months. This is likely a reflection of the more invasive nature of major hepatectomy when compared to percutaneous RFA [77].

RFA has been used in treating recurrence following surgical resection. One retrospective analysis of 40 patients with recurrent cholangiocarcinoma after hepatectomy investigated the role of RFA. Investigators found a median survival of 26.6 months and a 5-year survival of 18% [80]. Another single-center retrospective series examined 29 patients with recurrent cholangiocarcinoma following surgery treated for recurrence with RFA. The disease-free survival in this group at 4 years was 74%, the median survival after the RFA procedure was 27.4 months, and the overall 4-year survival was 21% [81]. Both studies had similar inclusion criteria including <3 lesions that were <5 cm. In the setting of tumor recurrence, patients are often not eligible for reoperation due to a variety of reasons including; comorbidities making surgery too risky, inadequate functional liver remnant, and anatomic considerations limiting resection. In these situations, limited data have suggested RFA as a reasonable substitution for resection when reoperation is contraindicated. In a prospective cohort study of 72 patients, Braunwarth et al. were able to increase the portion of patients with cholangiocarcinoma recurrence treated aggressively with curative intent from 12% to 37% by offering RFA to those ineligible or uninterested in hepatic resection. Though the majority of patients with recurrence underwent palliative care, those undergoing treatment with curative intent had significantly greater 5-year survival; 48% in the curative intent cohort vs. 12% in the palliative cohort [82]. The differences in 5-year survival between cohorts are likely affected by selection bias, however, these findings demonstrate how the less invasive nature of RFA allows for the treatment of additional patients deemed ineligible for surgery. 

### 5.3. Radiofrequency Ablation in Metastatic Disease to the Liver

Expert consensus guidelines indicate RFA for the treatment of metastatic disease in nonsurgical candidates with Childs Pugh class A or B with less than three metastases each less than 3cm in diameter [83,84,85]. Although there are no definite indications for metastatic disease, it is common practice to use the HCC guidelines for RFA treatment [85]. RFA for colorectal metastasis to the liver has demonstrated complete response rates of 52% to 95%. When used with non-curative intent, it can increase disease-free survival to 50% and overall survival to 94% at one year, however, the treatment is less effective with increasing lesion size [52]. In a single institution retrospective analysis of 194 patients with colorectal liver metastasis recurrence after hepatectomy, 50 underwent repeat hepatectomy and 144 underwent RFA. Indications for hepatectomy were anatomy that allowed for R0 resection, no surgical contraindication, adequate functional liver remnant, and appropriate hepatic inflow and outflow of the remaining liver. The indications for RFA were the number of tumors ≤3 and the maximum diameter of the tumor ≤5 cm, or the number of tumors >4, and the maximum diameter of the tumor ≤3 cm, a safe path for percutaneous ablation, the anatomy that allowed for R0 ablation, and no coagulopathy. The study found no significant differences in liver function or overall survival between the groups. The RFA group’s 5-year overall survival was 29%. Patients treated with RFA experienced lower rates of post-procedure complications and shorter procedure time and hospital length of stay [86]. Clearly, RFA has a role for patients to be treated with curative intent and may offer patients whose tumor characteristics prevent R0 resection a viable alternative. 

RFA and surgery have also been used synergistically in the treatment of metastasis to the liver. A retrospective single-institution study of 72 patients with liver metastases that were deemed unresectable underwent RFA in addition to surgery. Inclusion criteria were unresectable liver metastasis, the liver function of at least Childs Pugh A or B, and no extrahepatic involvement. The 5-year survival was found to be 19% and the 5-year recurrence-free survival of 13% [87]. This study included metastases of more lethal primary tumors including lung and pancreas which likely contributed to low overall survival. 

Pancreatic ductal adenocarcinoma (PDAC) patients with stage IV disease historically are treated with chemotherapy, however, RFA may offer extended survival in these patients who have liver metastasis. A retrospective single-institution study of 34 patients who had PDAC with liver metastasis and a diameter of the tumor ≤3 cm, less than five lesions, and no extrahepatic metastasis underwent RFA either intraoperatively or following pancreatectomy. Of patients that underwent RFA, 91% had a metastatic recurrence. However, 58% of patients with metastases had recurrence limited to the liver and 89% of those patients underwent repeat RFA. Again, 88% of patients that had a second RFA treatment had metastatic recurrence with 43% isolated to the liver. This group of isolated liver metastases underwent a third RFA treatment. Patients had a median survival of 14 months after liver metastases, however, there was no benefit in survival >2 years after surgery [88]. RFA may offer some benefit to patients with PDAC metastasis to the liver albeit in the setting of an aggressive primary cancer the benefits of treating liver metastases are inherently limited by the course of primary cancer.

Aside from prolonging survival, RFA treatment may offer a safe means of symptom relief for patients with hepatic neuroendocrine metastases. In a single institution prospective study of 80 RFA treatments for 63 patients with metastatic neuroendocrine tumors, Mazzaglia et al. found that 92% of participants experienced at least partial symptom relief while symptoms entirely resolved in 70% of cases. Only 5% of patients experienced perioperative morbidity and no 30-day mortalities were observed [67]. In symptomatic patients that otherwise have limited treatment options, RFA provided safe symptom and disease control. Additionally, recent early evidence suggests the modality may offer prolonged relief from malignant biliary obstruction when applied intraluminal. One small recent randomized controlled trial of 30 patients compared biliary obstruction between patients treated with placement of plastic biliary stent with intraductal radiofrequency ablation of malignant hilar biliary obstruction vs. placement of the plastic stent alone. For a subset of patients with ductal obstruction greater than 11 cm in length, intraductal RFA prolonged the median time to obstruction to 178 days vs. 122 days with plastic stenting alone [89]. A large, recent meta-analysis of 19 studies including 1946 patients demonstrates similar benefits for prolonging stent patency times as well as a slight overall survival benefit for intraductal RFA [90].

## 6. Microwave Ablation

### 6.1. Technique

Microwave ablation (MWA) emits electromagnetic radiation locally into tissue at frequencies between radiowaves and infrared radiation [15]. An oscillating electromagnetic field is created by a generator, passed through a power distribution system, and delivered to tissues through an antenna [15,16]. As with RFA, CT or ultrasound guidance is used to place the antenna within the tissue to be ablated (Figure 2). Tissues adjacent to the antenna are exposed to an electromagnetic field that oscillates at a high frequency. Molecules with dipole moments, such as water, experience an electromagnetic force as the dipole moment of the molecule attempts to align with the applied electromagnetic field. In the case of water, this may occur 2 to 5 billion times per second [15]. High-frequency alternation of the applied electromagnetic field generates heat in the nearby tissues as water molecules rotate and interact, generating heat. This heat reaches an excess of 100 degrees Celsius [16]. A cooling system exists in the antenna via chilled saline or gas [15,16].

Electromagnetic radiation emitted from a point source follows the inverse square law whereby the intensity of the electromagnetic field emitted by the antenna, and thus heat energy created within adjacent tissues, decreases precipitously at increased distances from the source. The field of ablation can vary based on the characteristics of the energy source, the shape of the electromagnetic field generated by the antenna, and the duration the tissue is exposed to the microwave. Given the ubiquity of water throughout the human body, microwaves are able to generate heat within diverse types of tissue including those with poor conductivity, high impedance, and low thermal conductivity which may limit other ablative therapies such as RFA [16]. Potential advantages to microwave ablation are higher temperatures achieved within the tumor, less dispersion of heat due to the heat sink effect, larger tumors amenable to ablation, improved convection of heat, and shorter procedure times [15,16]. Disadvantages include higher cost, less definitive visualization intra-procedurally during active ablation, underpowered systems, the large diameter of the antenna, and long thin ablation zones [16].

### 6.2. Microwave Ablation in Non-HCC Primary Liver Malignancy

Data describing clinical outcomes of MWA in cholangiocarcinoma are limited. In a prospective series by Yu, 15 patients were treated with MWA for cholangiocarcinoma. These patients were not surgical candidates and three patients had failed other treatment modalities. Additional criteria included less than three total lesions, each less than 5 cm, no portal venous thrombus or extrahepatic metastasis, and no coagulopathy. Median survival was 10 months, and the 1-year overall survival was 60% [91].

MWA offers several technical advantages over RFA leading to widespread adoption in modern interventional practices. MWA provides a more predictable ablation zone, allows for the treatment of multiple lesions at the same time, and achieves larger coagulation volumes while having a shorter procedural time [92]. Whether these attributes translate to better outcomes when applied to cholangiocarcinoma is poorly understood. A retrospective review compared RFA to MWA for the treatment of primary cholangiocarcinoma and intrahepatic metastases in 20 nonsurgical candidates, 76% of the tumors treated were metastasis from a primary cholangiocarcinoma. Tumors were less than 5 cm, and they did not have hilar cholangiocarcinoma or metastasis from hilar cholangiocarcinoma. Median disease-free survival was 8.2 months and median overall survival was 23.6 months for all 20 patients. No difference in tumor progression was observed between tumors treated with RFA and MWA [93].

A retrospective single institution series compared repeat hepatectomy to thermal ablation, which included both RFA and MWA in patients that had ≤5 lesions that were ≤5 cm in diameter, Child−Pugh A or B liver function, no coagulopathy, no evidence of vascular invasion or extrahepatic metastasis. The criteria for repeat hepatectomy was a single tumor or 2–3 tumors ≤ 5cm in diameter and otherwise the same criteria as thermal ablative therapy. No significant difference in terms of median survival or 3-year survival was observed between the two groups. Median overall survival for hepatectomy was 20.3 months and thermal ablative therapy was 21.3 months. The 3-year survival for hepatectomy was 17% and thermal ablative therapy was 21% [94]. 

### 6.3. Microwave Ablation in Metastatic Disease to the Liver

In the US, the most common metastasis to the liver is colorectal cancer, the treatment of which is the most commonly studied application of MWA for metastatic disease to the liver. A meta-analysis of 395 patients with colorectal metastases treated MWA demonstrated overall survival at 1, 3, and 5 years of 87%, 60%, and 45% and recurrence-free survival at 1, 3, and 5 years of 65%, 45%, and 34%, respectively. These outcomes lead the authors to conclude that MWA can safely be considered as an option for curative treatment of metastases <3 cm [95]. 

Similar to RFA, limited retrospective studies have compared outcomes between patients selected to undergo surgical resection and those treated with MWA due to one or more factors making the patient ineligible for resection. For example, Stattner et al. retrospectively reviewed 43 patients who had colorectal liver metastasis that was treated with MWA alone or MWA with surgery and found that there was a 36% 3-year survival for MWA alone vs. 45% for MWA with surgery [96]. Differences in outcomes between these groups are likely affected by selection bias, whereby patients with less severe disease are more often recommended standard of care treatment with resection while those with one or more factors making the patient ineligible for surgery are recommended MWA. When controlling for differences between treatment groups, MWA appears to be an equivalent option for the treatment of focal metastatic disease in the liver. A prospective randomized control trial of 30 patients with colorectal liver metastasis that were less than 8 cm, less than 10 lesions, and did not have cirrhosis, chronic hepatitis, or extrahepatic disease found no significant differences between surgery and MWA in mean survival times or 1-year, 2-year, and 3-year survival rates [97]. Treatment of colon cancer metastases is not limited to either surgery or percutaneous ablation; MWA can be administered via laparoscopic or open surgical approach. A retrospective single-institution series examined MWA in 36 patients with nonresectable colorectal metastasis to the liver. These patients underwent surgical ablation due to anatomical challenges that prevented the percutaneous approach. Intraoperative administration of MWA demonstrated a median overall survival of 81 months with a per-lesion recurrence rate of 4% [98]. 

Taken together these results demonstrate that MWA is a reasonable alternative to surgery and that in appropriate situations this ablative modality can function synergistically with the surgical approach in the treatment of colorectal liver metastasis. Whether these findings can be extrapolated to the treatment of liver metastases from other primary cancers requires further investigation. In the case of more aggressive primary cancers, treating metastases may not hold as much benefit to overall survival as with colorectal cancer.

## 7. Cryotherapy

### 7.1. Technique

The inverse of heat killing is cryoablation. Freezing temperatures between −20 and −190 degrees Celsius are achieved by liquid nitrogen or argon gas within an applicator probe causing an ice ball to form in tissues adjacent to the probe. Two freeze–thaw cycles are performed, which causes disruption of cell membranes, intracellular and extracellular ice formation, hypoxia, and ultimately cell death [17,18,19,20,21]. Guidance of the probe into the lesion and visualization of ice ball formation is obtained via CT, ultrasound, or MRI (Figure 3). This allows the physician to recognize the margin borders of the ablation in real time [17,18,21]. The ideal margins are 0.5 cm to 1 cm [19]. Additional probes may be inserted as necessary. Cryoablation can be performed percutaneously or intraoperatively during surgery. For example, if clean surgical margins are unobtainable, edge cryotherapy may reduce recurrence [99,100]. Similar to RFA, cryoablation is also limited by the heat sink phenomenon in which the convection of heat from nearby blood flow prevents the applicator probe from achieving the desired temperatures [17,21]. 

### 7.2. Cryoablation in Non-HCC Primary Malignancy and Metastatic Disease to the Liver

Limited studies describe the role of cryoablation in the treatment of non-HCC liver malignancies. Of those available, most describe the application of this modality intraoperatively for the treatment of HCC and metastasis, very little information is available in the literature describing the percutaneous use of cryoablation or the use of this modality in the treatment of cholangiocarcinoma. A single institution retrospective cohort study examined percutaneous, image-guided cryoablation in 186 patients with 299 primary and metastatic tumors within the liver that were deemed inoperable. Of the 56 primary tumors treated by cryoablation, 50 of these tumors were HCC, a remaining six patients were cholangiocarcinoma. The vast majority of tumors, 243 in total, were metastatic diseases from at least 17 different types of primary tumors. The highest technical efficacy rate was for renal cell carcinoma metastasis at 100% and the lowest was for lung cancer at 63%. Tumor progression was more likely in lesions larger than 4 cm and occurred in 23% of the lesions overall. CT guidance was associated with better technical success and lower rates of local tumor progression when compared to MRI guidance. Overall, the adverse event rate was 34% [101]. Interestingly, renal cell carcinoma metastasis appeared to respond well to cryoablation in the liver, this modality has become the standard of care in treating primary renal cell carcinoma. A limitation of this study is that the patients were taken from 16 years of experience with the procedure, however, half of the patients were treated in the first year of the study when the technique was first brought to this institution. This likely decreased the efficacy in those patients who were treated early as the operator was becoming comfortable with the technique. Unfortunately, there was no survival data presented specifically for cholangiocarcinoma patients. A lack of survival data remains a gap in the current literature regarding cholangiocarcinoma treated by cryoablation [102]. 

A prospective series of 223 patients who had colorectal cancer with liver metastasis observed outcomes in 168 patients who underwent liver resection alone and 55 patients who were treated with intraoperative cryotherapy, of which 25 were in combination with liver resection. Cryotherapy was used when the surgeons felt that the lesions were unresectable and if there was no extrahepatic disease. Cryotherapy was performed laparoscopically or after a bilateral subcostal incision. The groups were different in that the cryotherapy group had more patients who had a previous liver resection, the lesions were smaller, and they were less frequently synchronous. They found that the median survival time was 29 months in both groups and 5-year overall survival was 23% in surgical resection and 26% in cryotherapy. They also found that cryotherapy had 11% morbidity and 2% mortality while surgery had 26% morbidity and 5% mortality although there was no statistical significance reached for mortality. The median disease-free survival times and 5-year disease-free survival rates following resection were 10 months and 19% whereas cryotherapy was 6 months and 12%. The disease-free interval in the liver was 19 months and a 5-year liver disease-free rate of 33% after surgical resection, whereas cryotherapy was 7 months disease-free interval in the liver and a 15% 5-year liver disease-free rate [103]. Although surgery is the current standard, cryotherapy has similar results. Furthermore, cryotherapy produces less morbidity and mortality and may be preferred in more frail patients. 

A randomized clinical trial of 123 patients compared the cryosurgical technique to conventional surgical resection of metastasis from all forms of the primary malignancy. The 3-year survival rate in cryotherapy was 60% and 51% in surgical resection. The 5-year survival rate was 44% in cryotherapy and 36% in surgical resection. The 10-year survival rate was 19% in cryotherapy and 8% in surgical resection. Recurrence in the liver was observed in 85% of cryotherapy and 95% in surgical resection [104]. A Cochrane review of this study gave the evidence a grade of low certainty and raised concerns of bias and noted multiple statistics of interest that were not included such as time to mortality, cancer mortality, health-related quality of life, and time to progression of liver metastasis [105].

## 8. Irreversible Electroporation

### 8.1. Technique

Irreversible Electroporation (IRE) is a nonthermal technique that triggers programmed cell death by disruption of cellular membranes via applied a DC current with high electric potential. This ablative method requires the placement of multiple electrodes connected to a generator between the targeted tumor. The probes are placed percutaneously with ultrasound or CT guidance. A faraday probe and oscilloscope are connected to a computer which allows the monitoring of the current as it is applied. The patient is attached to EKG leads for anesthesia monitoring as well as cardiac gating. This technique creates high-voltage electrical pulsations of short duration that cause small pores within cell membranes, loss of contents, and programmed cell death via apoptosis [17,21,22,23]. The cells are affected, however, the extracellular matrix is not. This allows for the sparing of fragile structures immediately adjacent to, but not between the electrodes such as vessels and bile ducts [21,106,107]. There is concern that the closer the target lesion is to the heart, the more likely IRE will cause dysrhythmias. Therefore, heart-gated delivery is required to avoid this complication [22]. 

### 8.2. Irreversible Electroporation in Non-HCC Primary Liver Malignancy and Metastatic Disease to the Liver

IRE is a relatively new ablative modality that lacks clinical evidence when compared to incumbent ablative technologies [108]. A meta-analysis of IRE in liver cancer identified nine studies with 300 total patients. The vast majority of these patients, 123 in total, were treated for HCC. Only 21 were treated for cholangiocarcinoma. The remaining 156 patients had metastatic disease or disease classified as other [109]. All studies in the meta-analysis were observational in nature and most were retrospective, which limits the ability to draw conclusions on its effectiveness. One prospective cohort study found 6-month local recurrence of IRE-treated tumors to be 33% in cholangiocarcinoma, 38% in colorectal metastasis, and 75% in unspecified metastasis [110]. A more recent prospective trial demonstrated median overall survival of 21 months in 12 patients treated for cholangiocarcinoma who were deemed inappropriate for surgical resection or radiotherapy ablation [111]. 

Despite limitations in understanding the effectiveness of IRE, the included studies were able to describe the application, potential value, and imaging findings associated with the use of IRE in liver malignancy. For example, multiple studies demonstrated that when properly placed, IRE was able to avoid biliary injury [107,112]. Several others described characteristic post-ablation CT/MRI findings [113,114]. Another evaluated safety, short-term complications, and morbidity at a six-month follow-up [115]. A recent prospective trial demonstrated a major adverse event rate of 50% following the procedure [111]. While these studies describe the application and risks of IRE, additional studies should address the lack of randomized, prospective clinical outcomes in the use of non-HCC liver malignancy.

## 9. High-Intensity Focused Ultrasound

### 9.1. Technique

High-intensity focused ultrasound (HIFU) is a noninvasive technique that creates both thermal and mechanical effects to ablate tumors through coagulation necrosis. HIFU works similarly to a conventional ultrasound, however, the intensity is vastly greater. Conventional ultrasound has an intensity of 720 mW/cm^2^ whereas HIFU has an intensity of up to 10,000 mW/cm^2^ [24]. HIFU can be constant and cause thermal damage to cells or pulsed and cause acoustic cavitation. When HIFU is constant, target temperatures reach greater than 56 Celsius to allow for cell death. When HIFU is pulsed with alternating compression and expansion acoustic pulses, the target tissue has rapid pressure changes at points of weakness in the tissue creating gas and vapor-filled cavitation. When the cavities oscillate, mechanical shearing allows for cellular destruction [24,25,26]. The procedure is conducted with MRI or ultrasound guidance with no radiation or dose limit. A gel or liquid coupling device is placed in between the probe and target tissue to allow for the propagation of the ultrasonic waves [24,116]. These probes are extracorporeal, interstitial, or transrectal [24,116]. A potential limitation for treatment in the liver is the ability of HIFU to effectively ablate tumors deep in the parenchyma, obstruction by ribs, and organ movement throughout the procedure [117].

### 9.2. High-Intensity Focused Ultrasound in Non-HCC Primary Liver Malignancy and Metastatic Disease to the Liver

HIFU is an emerging ablative therapy for the treatment of various malignancies, most notably, prostate in the US. Most clinical studies of HIFU applied to liver malignancy focus primarily on HCC. Recent studies have investigated the feasibility and safety of HIFU in combination with other therapies, such as transarterial chemoembolization (TACE) in the treatment of HCC. One study of 37 patients demonstrated a 2-year progression-free survival of 29.7% and a median overall survival of 24 months [118]. Similar studies have been performed in non-HCC liver malignancies. One cohort study of 12 patients with hepatoblastoma examined HIFU in combination with TACE. The authors found a median survival time of 14 months and a 2-year survival rate of 83% [119]. A systematic review identified 24 separate studies from 2008 to 2019 that described the technical application of HIFU in 940 patients: 924 had HCC, 12 had hepatoblastoma, and 4 had cholangiocarcinoma [120]. Results were not differentiated between tumor types. Complete tumor ablation was found in 55% of patients. When combined with other modalities such as transarterial chemoembolization, RFA, or Percutaneous Ethanol Injection (PEI), technical success increased to 66%. The most common complications observed were skin burns (15%), local pain (5%), and fever (2%). Though the study found promising rates of technical success, the review was limited by wide heterogeneity between methods of the included studies as well as a lack of clinical outcomes such as survival data for some studies. 

In regard to metastatic disease of the liver, Leslie et al. examined two phase II non-randomized prospective trials of thirty-one patients in total (one was excluded due to equipment failure), twenty-nine of whom had metastatic disease to the liver and one of whom had primary HCC. All patients had a radiological evaluation at 30 days. The radiological trial ended after this evaluation while the surgical trial went on to perform a resection of the tumor. Thirty days after HIFU treatment, 93% of patients demonstrated evidence of ablation on post-procedure imaging. Disease progressed in seven patients, and two died during the follow-up timeframe [117]. A larger, single-center retrospective analysis of 275 patients treated with HIFU compared technical success, disease progression, and survival between 80 patients with HCC and 195 with liver metastases. Interestingly, the study found similar rates of objective response (72% vs. 64%), disease control rate (81% vs. 83%), and a median one-year survival (13 vs. 12 months), suggesting HIFU may be similarly effective in the treatment of malignant liver lesions regardless of tumor type [121]. Overall, it is difficult to draw conclusions regarding the efficacy of HIFU without prospective, randomized clinical studies with longer follow up.

## 10. Photodynamic Therapy

### 10.1. Technique

Photodynamic therapy (PDT) is a technique that allows tumor ablation through a unique mechanism of photodamage. Systemic administration of a fluorescent photosensitizing agent localizes to the tumor and is activated by light of a specific wavelength creating a photodynamic reaction that creates oxygen-free radicals and induces cell death through apoptosis and necrosis [27,28,29]. Various forms of light are used including lasers, incandescent light, and laser-emitting diodes [122]. The light is directed through a catheter via a laparoscopic approach, with endoscopic retrograde cholangiopancreatography, through a T-tube, or with percutaneous placement into the interstitium of the tumor itself [123] (Figure 4). Limitations of PDT are poor tissue depth penetration of the light irradiation, possible revascularization of the tumor, and non-specificity of the photosensitizing agents [124].

### 10.2. Photodynamic Therapy in Non-HCC Primary Liver Malignancy

Compared to other ablative modalities, PDT is a more well-established treatment for Non-HCC primary malignancy. In a prospective cohort of 50 patients with hilar cholangiocarcinoma, patients were assigned one of three treatments based on multidisciplinary team discussion; radical surgery, PDT and stenting, or chemotherapy and stenting. Of the fifty patients, ten were surgical candidates, the remaining were placed into the PDT and chemotherapy groups for palliative therapy. To qualify for palliative therapy patients required an expected life expectancy of >6 weeks. Skin porphyria or extensive metastatic disease were contraindications to PDT. The study found that the actual 1-year survival was 80% in surgery, 75% in PDT and stenting, and 12% in chemotherapy and stenting. Furthermore, the median survival after 4 years of follow-up had not been reached in the surgery group, while the median survival was 169 days in the chemotherapy and stenting group, and 425 days in the PDT and stenting group [125]. Differences in survival at 4-year follow-up between PDT and surgery groups might relate to patients with less severe diseases being offered a surgical approach. Future prospective, randomized trials could directly compare the survival benefit of PDT vs. surgery, however, the survival demonstrated for PDT alone is promising. 

Randomized prospective trials demonstrate a similar survival benefit for PDT in another group of nonsurgical candidates. One randomized prospective multicenter study of 39 patients with Bismuth II-IV cholangiocarcinoma compared survival between endoscopic PDT with stenting across the biliary disease vs. stenting only in patients with no extrahepatic disease who were deemed inoperable. The authors found that the addition of PDT increased median survival to 493 days compared to 98 days without PDT. Patients who underwent PDT also had several improved metrics in their physical functioning status and quality of life when compared to those without PDT. Both groups had similar amounts of fatal and nonfatal adverse events [28]. Two separate systematic reviews discussing PDT in unresectable cholangiocarcinoma have found similar results [126,127]. 

Additional studies have investigated the role of combination therapies involving PDT. A retrospective cohort of 55 patients at a single center examined patients with unresectable cholangiocarcinoma who received PDT as well as other supplemental treatments such as chemotherapy alone, chemotherapy and radiotherapy, and radiotherapy alone. No significant difference was found in median survival comparing PDT alone vs. PDT and chemotherapy/radiotherapy [128]. Taken together these results demonstrate that PDT monotherapy offers nonsurgical candidates a one-year survival benefit comparable to surgery without the systemic side effects of other adjuvant therapies. 

### 10.3. Photodynamic Therapy in Metastatic Disease to the Liver

PDT is most commonly used via an endoscopic approach to treat malignancies adjacent to the biliary tree. As such, this modality is less commonly applied to the treatment of metastatic disease which may occur anywhere throughout the liver with poor penetration of endoscopic light source. A phase I trial examined the technical success and safety characteristics of percutaneous, image-guided PDT in the treatment of 24 patients with unresectable colorectal metastasis to the liver. These patients had previously resected primary colorectal carcinoma and also had no evidence of extrahepatic disease. Exclusion criteria included lesions >7 cm, patients who had poor functioning status, diseases that cause damage when exposed to light, coagulopathy, chronic liver disease or elevated LFTs, or prior treatment with chemotherapy, photosensitizing drugs, or radiotherapy 30 days before PDT. In terms of technical success, all lesions treated experienced necrosis at 1 month. Safety outcomes included; one serious adverse event of bleeding that recovered uneventfully, two moderate adverse events including pancreatic injury, and a skin lesion at the site of PDT fiber insertion which took several months to resolve [129]. With PDT showing promising results in cholangiocarcinoma, further investigation of its efficacy in metastatic liver disease should be considered.

## 11. Stereotactic Body Radiotherapy

### 11.1. Technique

Stereotactic body radiotherapy (SBRT) is given through a linear accelerator or a specific delivery device such as the robotic Cyberknife [30]. Commonly, the linear accelerator administers doses of 30 to 60 Gy in five fractions or fewer for metastasis, five or fewer, and tumors up to 6 cm [31]. Preprocedural imaging may be obtained with CT, MRI, or PET combinations. Multiple beams are used, and these beams are directed from many different angles. Corrections for movement are made through breath holding, abdominal compression devices, gating the radiation beams for respiratory variation, and placement of internal markers are used for tumor tracking [127,128,129]. Cyberknife is a commonly used robotic SBRT machine with an arm that allows for a 6-dimensional movement of a small linear accelerator. The tumor is localized with an X-ray and infrared tracking system that has both internal and external components that correct for any movement by the patient during the procedure [32]. A major drawback of SBRT is the associated toxicity, specifically for the liver it may cause radiation-induced liver injury [130].

### 11.2. Stereotactic Body Radiotherapy in Non-HCC Primary Liver Malignancy

SBRT has shown good survival in cholangiocarcinoma at highly specialized centers that perform the procedure. Jung conducted a retrospective single-institution study examining 58 patients who had unresectable primary cholangiocarcinoma or recurrent cholangiocarcinoma treated by SBRT. Of note, five of these patients were treated with external beam radiation therapy as well and were selected by radiation oncologist discretion. There was a median follow-up of 10 months. In the primary cholangiocarcinoma cohort, the 2-year overall survival rate was 11% and median survival was 5 months. In the recurrent cancer group, the median survival was 13 months and the 2-year survival was 28% [131]. A retrospective review of twenty-eight patients who had cholangiocarcinoma who underwent SBRT and did not have distant metastasis or ascites, had a maximum of three lesions, eight had previously been treated by transarterial chemoembolization, 8 patients had cirrhosis, and there were 20 Childs Pugh class A patients and eight Childs Pugh class B patients. They found a median overall survival of 15 months, 2-year survival rate was 32%, and the 2-year progression-free survival rate was 21% [132]. A single institution retrospective cohort of 31 patients with cholangiocarcinoma who underwent SBRT. They excluded patients with lesions >8 cm, active infection, their comorbidities prevented them from being a liver transplant candidate, or extrahepatic disease. They found that the median survival was nearly 16 months, 2-year overall survival was 33%, and the 2-year freedom from disease progression was 34%. They offered four patients liver transplantation after SBRT and the median overall survival was 31 months [133]. Certainly, SBRT should be part of the discussion for patients with cholangiocarcinoma. 

### 11.3. Stereotactic Body Radiotherapy in Metastatic Disease to the Liver

Similar to non-HCC primary liver malignancy, SBRT has shown good outcomes for patients compared to other ablative therapies. A prospective phase I and II clinical trials at a single center examined SBRT in colorectal metastasis. They included lesions of any size or amount in phase I and limited lesions to ≤15 cm or 5 lesions in phase II. Patients achieved a median overall survival of 16 months and 4-year overall survival was 9% [134]. A separate phase II clinical trial included 42 patients with colorectal metastasis to the liver who had contraindications to surgery as well as metastasis not amenable to RFA. They included patients who had no extrahepatic disease, no concurrent chemotherapy around the time of SBRT, maximum lesion size of 6 cm, ≤3 total lesions, no prior radiation to the targeted area, appropriate liver function and coagulation parameters, no connective tissue disorders, and adequate performance score. Patients had a median overall survival of 29 months and 24-month actuarial local control of the disease was achieved in 91% of patients [135]. A limitation of this study is that the follow-up of 2 years is shorter when compared to studies of more established treatments. The results are reassuring that patients who may otherwise only be considered for palliative treatment have an option for local control of their lesions and an increase in survival rate. A single institution retrospective study of 323 patients who underwent hepatectomy for colorectal metastasis, of which 206 developed recurrence. These recurrence patients underwent surgery, RFA, or SBRT. Surgery was the preferred option and RFA or SBRT was chosen if the patient was not a surgical candidate. RFA was performed in 1–3 lesions ≤3 cm and SBRT in 1–3 lesions ≤5 cm, with RFA being preferred. At 2 years, surgery had local disease control of 93%, SBRT of 74%, and RFA of 56% [136]. Although SBRT is not as efficacious as surgery, it may warrant more frequent use especially if ablative procedures are being considered. 

## 12. Laser-Induced Thermotherapy

### 12.1. Technique

Laser-induced thermotherapy (LITT) uses optical fibers to produce radiation that induces tumor coagulation necrosis at temperatures of 130 Celsius. MRI is the preferred imaging modality to visualize the lesion [33]. Periprocedural temperature monitoring of the applicators allows for customized treatment of each lesion [137]. LITT optical fibers are applied percutaneously, laparoscopically, or in an open-operative approach. The laser applicator has a built-in liquid cooling system that allows a large zone of necrosis when compared to other ablative modalities making it especially useful in the treatment of metastasis [138]. Multiple applicators may be inserted depending on the tumor size. For example, lesions of 2 cm are treated with two applicators and lesions of 3 cm are treated with three applicators. The laser power is applied and then the applicators are withdrawn 1–2 cm and this is repeated until the tumor is ablated [138]. A challenge of LITT is procedural time, which may take 1–2 h [137].

### 12.2. Laser Induced Thermotherapy in Non-HCC Primary Liver Malignancy and Metastatic Disease to the Liver

Similar to other less common ablative therapies, there are no studies of LITT in non-HCC primary malignancy. Most available data on LITT comes from the application of this modality in the treatment of metastatic disease at single institutions. Results of the effectiveness of LITT varied between institutions based on volume, experience, and inclusion criteria. One large prospective trial of 1259 patients with 3440 metastases from a variety of primary tumors treated over 14,694 laser applications demonstrated median survival of three years in patients with <5 lesions each less than 5 cm in diameter [138]. Puls et al. studied 87 nonsurgical candidates with colorectal metastases treated with 180 sessions of LITT at a single institution. The median survival was 54 months. The 1-year survival rate after the first ablation was 96% and the 5-year survival rate was 33% [137]. Another large prospective cohort of 594 nonsurgical candidates with colorectal liver metastasis following initial hepatectomy treated roughly half the patients with curative intent. They found a median survival of 25 months after the date of the first LITT procedure, a 1-year survival rate of 78%, and a 5-year survival rate of nearly 8%. Having more than four metastases decreased median survival to 18 months and patients with a metastasis size greater than 4 cm had a median survival of 21 months [139]. 

## 13. Electrochemotherapy

### 13.1. Technique

Electrochemotherapy leverages electroporation to increase the uptake of chemotherapy within a targeted lesion. In this method, chemotherapy can be provided intratumorally or intravenously, but uptake of the chemotherapy is particularly high within the targeted tumor in comparison to surrounding tissues. The procedure is performed with an electric pulse generator and multiple electrodes. The electrodes are placed inside and around the tumor. Chemotherapy is then given and afterward the tumor is exposed to the electric field causing increased permeability of the cells. During this process, the patient is monitored on EKG and electricity is administered during the refractory period of the cardiac cycle to prevent arrhythmias [34]. Historically, the limitation of electrochemotherapy in the treatment of liver malignancy was a requirement for open surgical technique. Recently this challenge has been overcome by the use of the percutaneous approach [140].

### 13.2. Electrochemotherapy in Non-HCC Primary Liver Malignancy and Metastatic Disease to the Liver

The use of electrochemotherapy in liver malignancy is relatively new. In a prospective cohort of five patients with unresectable cholangiocarcinoma electrochemotherapy was utilized in one patient intraoperatively while undergoing hepatectomy, three underwent percutaneous electrochemotherapy for a singular nodule, and the last patient underwent electrochemotherapy in conjunction with RFA to treat liver recurrence after hepatectomy. Responses ranged from death within 10 months due to cardiovascular failure to complete response at 18 month follow-up [141]. In a prospective pilot study of 16 patients with metastatic colorectal liver metastasis, electrochemotherapy was used intraoperatively. The inclusion criteria specified that the patient has a good performance status, life expectancy >3 months, age over 18, and a chemotherapy-free interval of 2–5 weeks. Not every lesion was treated with electrochemotherapy, and some lesions were only partially treated. Pathologic comparison of resected metastases in the same patient treated with and without electrochemotherapy demonstrated a larger percentage of the residual tumor when treated with resection alone. The study did not observe any perioperative mortality and or other serious adverse events related to the electrochemotherapy. Despite nearly half of the patients with metastatic disease near a major blood vessel, no intraoperative or postoperative bleeding occurred. The median radiologic follow-up interval was 33 days and 85% of patients had a complete response [142]. A follow- up study at 8–10 weeks examined the histological changes of the tumors treated with electrochemotherapy and found regressive changes in the area of the liver with the tumor while preserving blood vessels >5 mm and coagulating smaller vessels [143]. While these initial findings are promising, further research is needed to determine the safety and efficacy of this modality in liver malignancy in larger cohorts outside of colorectal metastatic disease. 

## 14. Percutaneous Ethanol Injection

### 14.1. Technique

Using a long needle, ethanol or acetic acid is injected percutaneously into the tumor under CT or ultrasound visualization. The injection results in coagulative necrosis of the tumor [35]. Patients undergo multiple treatments with follow-up imaging to observe tumor response [144]. Compared to other ablative modalities percutaneous ethanol injection can be quite tedious and laborious which has hindered the widespread adoption of this modality. 

### 14.2. Percutaneous Ethanol Injection in Non-HCC Primary Liver Malignancy and Metastatic Disease to the Liver

Studies regarding percutaneous ETOH injection in primary malignancy have exclusively focused on HCC. A study of 14 patients who underwent percutaneous ethanol injection in metastatic tumors of any primary malignancy included patients without extrahepatic disease, <3 lesions, <4 cm size of the biggest lesion, refusal of surgery when offered, no other anticancer therapies, and no coagulopathy. A complete technical response was observed in 11 lesions and the maximum recurrence-free follow-up was 38 months. Technical success was more likely in lesions with a diameter of less than 3 cm [145]. Further studies are needed understand the role of ethanol injection in treatment of liver metastasis. Even though ethanol was more commonly used for treatment of primary HCC, a Cochrane review found insufficient evidence in the literature demonstrating that percutaneous ethanol injection performed better than no intervention [146]. 

## 15. Conclusions and Future Directions

Ablative therapies have a long history of treating various pathologies, however, the evidence supporting their use in liver malignancy is still under development. With regard to the treatment of liver malignancies, the most readily available data pertains to primary HCC rather than metastatic disease or non-HCC primary malignancy. Ablative therapy indications have been proposed for metastatic disease to guide patient and procedure selection [147]. The use of ablative therapy in non-HCC primary liver malignancy and metastatic malignancy has shown many viable treatment options for unresectable lesions. In certain patients, RFA and MWA may be considered curative. Ablative therapies offer good options for palliative treatment. Surgeons should continue to consider the addition of ablative therapy to patients undergoing resection. The study of ablative modalities in non-HCC liver malignancy is complicated by ever-changing ablative technologies and differences between protocols of single-institution observational studies. Heterogeneity in the natural progression of primary malignancy has bearing on survival data when evaluating the potential benefit of ablating liver metastases. In the case of cholangiocarcinoma, low incidence and poor survival at the time of diagnosis limit the number of patients available to evaluate ablative techniques. Finally, the availability of surgical resection for ideal surgical candidates introduces selection bias which must be considered when evaluating outcomes of ablation in patients with comorbidities not eligible for surgery. 

Even still, a few modalities have demonstrated promising results in limited prospective, randomized trials, such as PDT in the treatment of cholangiocarcinoma or MWA in the treatment of solitary colorectal metastasis. Based on available evidence, certain percutaneous ablative modalities are reasonable considerations when surgical resection is not possible. A further prospective investigation of ablative technologies is warranted to better evaluate these technologies prior to widespread adoption, and investigators should examine ablative therapies’ efficacy prospectively, ideally in randomized control trials, and develop new modifications to improve their use in the future. However, the need for further research supporting ablative methods will have to compete with widespread, growing interest in modern catheter-directed transarterial chemoembolization and radioembolization with promising results demonstrated in randomized trials of their own [148]. Finally, improvements in liver transplant surgery have allowed some investigators to revisit dogma about the inappropriateness of liver transplantation for the treatment of non-HCC liver malignancy. Outcomes from multi-center, prospective trials have demonstrated impressive results for liver transplantation as treatment for both unresectable colorectalcarcinoma metastases and intrahepatic cholangiocarcinoma. 

## Figures and Tables

**Figure 1 cancers-15-01200-f001:**
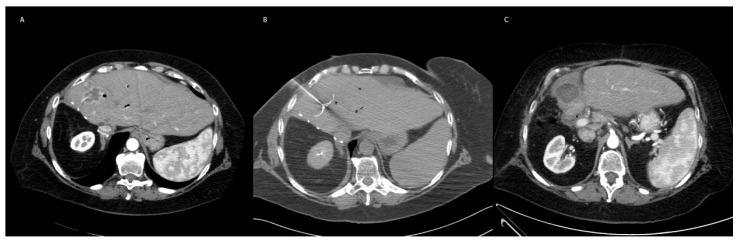
Representative Image of RFA. (**A**) 56-year-old female with hilar cholangiocarcinoma status post right hepatectomy and extra hepatic bile duct resection followed by multiple embolizations for residual disease. Referred for RFA ablation for treatment of enlarging segment 4A and 4B metastases in the setting of hepatic artery stenosis limiting repeat embolization. The preprocedural axial CT demonstrates a contrast enhancing hypoattenuating cholangiocarcinoma in segment 4 adjacent to the hepatectomy margin. (**B**) Intraoperative CT demonstrating percutaneous radiofrequency probe (Boston Scientific LeVeen) within the segment 4A liver mass. Delivered 190 watts for 15 min followed by 200 watts for 15 min without reaching impedance limit. (**C**) Axial CT one month following the procedure demonstrating a hypoattenuating lesion with thin rim of contrast enhancement overlying the zone of ablation.

**Figure 2 cancers-15-01200-f002:**
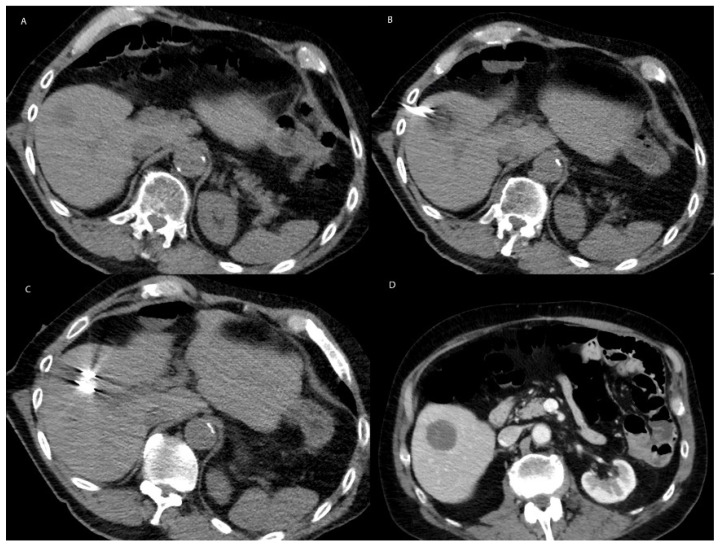
Representative image of MWA. (**A**)Noncontrast enhanced preprocedural axial CT demonstrating a well circumscribed hypoattenuating lesion in segment 5 concerning for metastasis in a 74-year-old male with rectal carcinoma. (**B**) The lesion was ablated by application of 100 watts for 7 minutes through each of the adjacent probes. The ablation zone is visualized as a region of hypo attenuation along the antennae. (**C**) Under CT guidance, two Neurowave microwave antenna are positioned within the lesion to provide overlapping ablation zones. (**D**) Contrast enhanced axial CT demonstrating a hypoattenuating lesion overlying the previous ablation zone.

**Figure 3 cancers-15-01200-f003:**
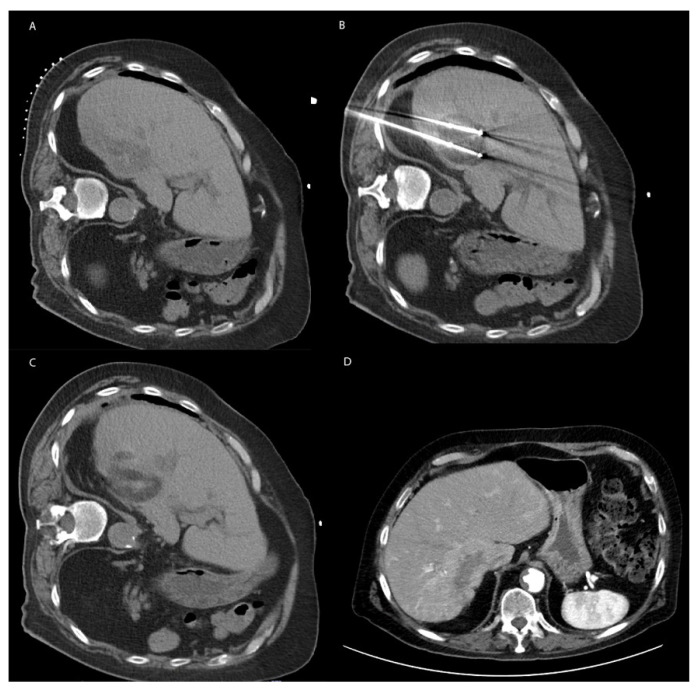
Representative image of Cryotherapy. (**A**) Noncontrast enhanced axial CT demonstrating a large hypoattenuating metastasis in liver segment 7 of an 83-year-old male with history of ascending colon cancer status post hemicolectomy. (**B**) Intraoperative noncontrast axial CT demonstrating ice ball formation around two percutaneous cryoablation probes within the metastasis. (**C**) Immediately following cryoablation, thermal damage is demonstrated by a region of hypoattenuation. (**D**) Two years later, the patient unfortunately presented with another large metastatic lesion within right liver lobe concerning for recurrence. Deemed unresectable by Surgery and too large for repeat cryoablation.

**Figure 4 cancers-15-01200-f004:**
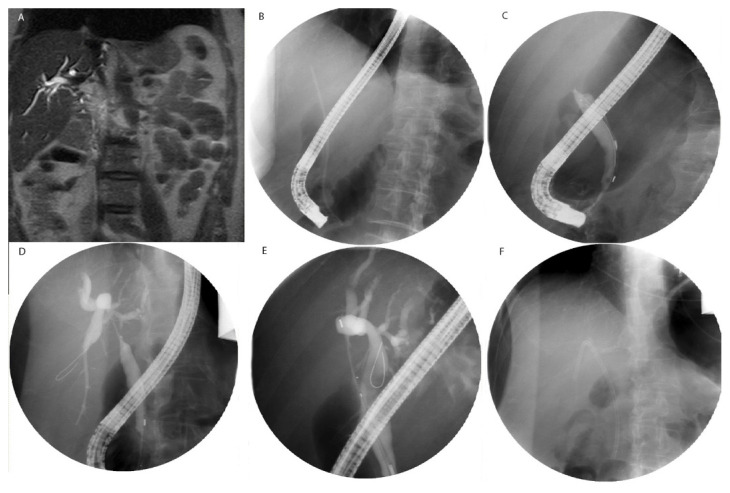
Representative image of PDT. (**A**) Sagital MR Abdomen T2 weighted image demonstrating a soft tissue perihilar mass and diffuse dilation of the intrahepatic biliary system in a 70-year-old male diagnosed with choliangiocarcinoma. (**B**) Initial flouroscopic image during ERCP demonstrating position of the scope and an intrahepatic biliary stent. (**C**) Following removal of the stent, contrast injection into the ampulla of Vater demonstrates common bile duct stricture. (**D**) After crossing the common bile duct stricture, injection of contrast media demonstrates the extent of focal common bile duct stricture and biliary dilation upstream. (**E**) Light diffuser is advanced over the stricture for application of PDT. (**F**) Following PDT session, the light diffuser is removed from the common bile duct and an endobiliary stent is placed.

**Table 1 cancers-15-01200-t001:** Summary of Ablative Therapy Techniques.

Ablative Therapy	Technique
Radiofrequency Ablation	A probe delivers electric current, which is alternated by a radiofrequency generator creating heat [6,13,14]
Microwave Ablation	A generator oscillates an electromagnetic field through an antenna, which causes molecules with dipole moments to feel electromagnetic force. The movement of molecules creates heat [15,16]
Cryotherapy	An applicator delivers argon gas or liquid nitrogen causing an ice ball to form adjacent to the probe [17,18,19,20,21]
Irreversible Electroporation	Electrodes are inserted into the target tissue and a high-voltage electric impulses created by the electrodes produce pores in the cellular membrane [17,21,22,23]
High-Intensity Frequency Ultrasound	A probe, similar to that which is used in conventional ultrasound, delivers ultrasonic waves at extremely high-frequency. These waves create pressure changes that cause gas or vapor-filled cavitations in the tissue. The cavitations oscillate and allow for the mechanical destruction of target tissue [24,25,26]
Photodynamic Therapy	Systemic administration of a fluorescent photosensitizing agent localizes to the tumor. A probe that emits light at a desired wavelength is directed toward the tumor. Activation of the agent by a specific wavelength of light creates free radical damage [27,28,29]
Stereotactic Body Radiotherapy	A linear accelerator or specialized device such as the Cyberknife delivers beams of radiation. A high degree of precision is achieved through patient immobilization, imaging, and flexible external placement of the radiation device [30,31,32]
Laser Induced Thermotherapy	Optical fiber applicators deliver light, which is absorbed by the target tissue creating heat [33]
Electrochemotherapy	Electrodes are inserted into the target tissue and chemotherapy is given systemically. An electric field is then created by the electrodes causing cells to become more porous and allowing for increased uptake of chemotherapy [34]
Percutaneous Ethanol Injection	A long needle is inserted and ethanol is injected into the target tissue [35]

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
