# Peer review of "Ablative Therapy in Non-HCC Liver Malignancy"

_cancers, 2023, doi:10.3390/cancers15041200_

Round 1

Reviewer 1 Report

First, I want to congratulate the authors on this well written paper. In the last few decades, it has become increasingly more difficult to navigate through all different types of interventional radiology tumor treatments. Implementing a comprehensive review such as this one, will provide important knowledge on what are our current options for treatment of liver metastasis and non-HCC primary liver tumors.

I would like to make some comments to potentially improve this paper:

1.     In extensive reviews like this one, the authors usually describe the pattern of their research and how they decide to include or exclude certain papers. The topic is extremely vast, and it will be interesting to know how you selected the information that you included in your manuscript. Did you use AI such as IRIS?  You can add this description under material and methods

2.      Under section 4 Impact of Metastatic Disease to the Liver on Prognosisin the discussion of colo-rectal metastasis you should mention the SECA I and SECA II prospective studies where the survival for nonresectable Mts is 83% at 5y after liver transplantation. In section 3 – same good result was obtained in hilar cholangiocarcinoma where the OLT was found to be more effective than resection.

Author Response

We have added a materials and methods section. We have added the sources you have suggested which are #18,29,30.

Reviewer 2 Report

This article reviews the available ablative therapy for the treatment of non-hepatocellular carcinoma liver malignancy. It is a very specific review article, summarizing in a comprehensive way the scarce available literature regarding the topic. No other articles have been published that review in such a deep way the ablative therapy in liver cancer, with examples and a good Discussion. The conclusions are consistent with the evidence and arguments presented and they addressed the main question posed.

It is a review article, and maybe they could cite more paper from 2022 if existing. Images are really good. A graphical abstract could be added.

Author Response

We have added a graphical abstract. We have added new sources which are #18,29,30,69,70,100,111. Sources 69,70,100, and 111 are from 2022.
